# Cytology and HPV Co-Testing for Detection of Vaginal Intraepithelial Neoplasia: A Retrospective Study

**DOI:** 10.3390/cancers15184633

**Published:** 2023-09-19

**Authors:** Frederik A. Stuebs, Anna K. Dietl, Martin C. Koch, Werner Adler, Carol Immanuel Geppert, Arndt Hartmann, Antje Knöll, Grit Mehlhorn, Matthias W. Beckmann, Carla E. Schulmeyer, Felix Heindl, Julius Emons, Anja Seibold, Annika S. Behrens, Paul Gass

**Affiliations:** 1Department of Gynecology and Obstetrics, Comprehensive Cancer Center Erlangen–European Metropolitan Area of Nuremberg (CCC ER-EMN), Erlangen University Hospital, Friedrich-Alexander-Universität Erlangen-Nürnberg, Universitaetsstrasse 21–23, 91054 Erlangen, Germany; 2Department of Gynecology and Obstetrics, Hospital ANregiomed Ansbach, Escherichstrasse 1, 91522 Ansbach, Germany; martin.koch@anregiomed.de; 3Department of Medical Informatics, Biometry and Epidemiology, Friedrich-Alexander-Universität Erlangen-Nürnberg, Waldstrasse 6, 91054 Erlangen, Germany; 4Comprehensive Cancer Center Erlangen–European Metropolitan Area of Nuremberg (CCC ER-EMN), Institute of Pathology, Erlangen University Hospital, Friedrich-Alexander-Universität Erlangen-Nürnberg, Krankenhausstrasse 8–10, 91054 Erlangen, Germany; 5Institute of Clinical and Molecular Virology, Erlangen University Hospital, Friedrich-Alexander-Universität Erlangen-Nürnberg, Schlossgarten 4, 91054 Erlangen, Germany; 6Gynecology Consultancy Practice, German Cancer Society (DKG) and Committee on Cervical Pathology and Colposcopy (AG-CPC) Certified Gynecological Dysplasia Consultancy Practice, Frauenarztpraxis Erlangen, 91054 Erlangen, Germany

**Keywords:** vaginal intraepithelial neoplasia (VaIN), vaginal cancer, colposcopy, cytology, pap smear, hysterectomy, human papillomavirus

## Abstract

**Simple Summary:**

Vaginal intraepithelial neoplasia (VaIN) is a rare premalignant disease, with an incidence ranging from 0.2 to 2 per 100,000 women per year. Although the risk factors for VaIN and cervical intraepithelial neoplasia (CIN) are similar, the incidence of VaIN appears to be 100 times lower. The correct diagnosis of VaIN remains challenging even for experienced examiners. Almost half of VaIN III/vHSIL ist associated with past cervical dypslasia. In the past decade, the diagnosis of VaIN has increased steadily. Cytology is valid tool for the diagnosis of intraepithelial neoplasia and carcinoma of the cervix and vagina. Only very limited data are available on the correct diagnosis of VaIN and vaginal cancer using cytology. In this retrospective study, we aimed to investigate the accuracy of cytology and HPV co-testing for the detection of VaIN, as these are often the first signs of VaIN: positive HPV testing during colposcopy increased the likelihood for VaIN II/III/vHSIL threefold.

**Abstract:**

(1) Background: Vaginal intraepithelial neoplasia (VaIN) is a rare premalignant disease caused by persistent human papillomavirus (HPV) infection. Diagnosing VaIN is challenging; abnormal cytology and positive HPV tests are usually the first signs, but published data on their accuracy for detecting it are rare and contradictory. The aim of this study is to compare the results of hrHPV and cytology co-testing with the histological findings of the vagina. (2) Methods: In the certified Dysplasia Unit at Erlangen University Hospital, cytology and HPV samples from the uterine cervix or vaginal wall after hysterectomy were obtained between 2015 and 2023 and correlated with histological findings in biopsies from the vaginal wall. Women without vaginal biopsy findings or concomitant cervical disease were excluded. (3) Results: In all, 279 colposcopies in 209 women were included. The histological results were: benign (*n* = 86), VaIN I/vLSIL (*n* = 116), VaIN II/vHSIL (*n* = 41), VaIN III/vHSIL (*n* = 33), and carcinoma (*n* = 3). Accuracy for detecting VaIN was higher in women with previous hysterectomies. Positive HPV testing during colposcopy increased the likelihood for VaIN II/III/vHSIL threefold. The detection rate for VaIN III/vHSIL was 50% after hysterectomy and 36.4% without hysterectomy. (4) Conclusions: Women with risk factors for VaIN, including HPV-16 infection or prior HPV-related disease, need careful work-up of the entire vaginal wall. Hysterectomy for HPV-related disease and a history of cervical intraepithelial neoplasia (CIN) also increased the risk for VaIN II/III/vHSIL.

## 1. Introduction

Vaginal intraepithelial neoplasia (VaIN) is a rare premalignant disease, with an incidence ranging from 0.2 to 2 per 100,000 women per year [1,2]. However, the incidence of the condition has steadily increased in recent years as a result of the widespread application of cytology and high-risk human papillomavirus (hrHPV) co-testing and colposcopy in cervical cancer screening [3]. Persistent infection with human papillomavirus (HPV) is considered to be the main risk factor for VaIN [4]. Additional risk factors that increase the likelihood of HPV infection are: multiple sexual partners, young age at sexual debut, and immunosuppressive treatment [1,5,6,7]. Prior or synchronous cervical intraepithelial neoplasia (CIN) or a history of hysterectomy for HPV-related CIN or invasive cervical cancer are also believed to be involved in the development of VaIN [8,9]. A higher risk of progression to vaginal cancer was reported in women diagnosed with VaIN III/vHSIL and for women with hysterectomy for HPV-related CIN [10].

As with CIN, VaIN is classified in accordance with the World Health Organization Classification of Female Genital Tumors: VaIN I lesions can be classed as low-grade squamous intraepithelial lesions (LSILs), whereas VaIN II and VaIN III both represent high-grade squamous intraepithelial lesions (HSILs), depending on the depth of the involved tissue [4,11,12]. According to the Lower Anogenital Squamous Terminology (LAST), the recommended terminology for HPV-associated squamous lesions of the LAT is LSIL and HSIL, which may be further classified by the applicable—IN subcategorization (e.g., vagina = VaIN 3) [13]. LSILs are regarded as proliferations of squamous or metaplastic cells with abnormal nuclear features, including increased nuclear size, irregular nuclear membranes, and increased nuclear-to-cytoplasmic ratios [13]. LSIL is a common finding, and self-limited HPV infections will resolve spontaneously [13]. HSIL is defined by the proliferation of squamous or metaplastic squamous cells with abnormal nuclear features, including increased nuclear size, irregular nuclear membranes, and increased nuclear-to-cytoplasmic ratios accompanied by mitotic figures [13]. HSILs can progress to invasive cancer if untreated [14].

Although the risk factors for VaIN and CIN are similar, the incidence of VaIN appears to be 100 times lower [1,9]. One possible explanation is the absence of a vulnerable squamocolumnar junction in the vagina [15]. The frequency of HPV infections in the vagina is believed to be similar to that in the cervix, but a lytic cellular reaction in the vaginal epithelium allows regression of lesions, in contrast to the latent infections observed in the cervix, which cause persistent dysplasia [1,16,17]. The prevalence rates of HPV in vaginal cancer, VaIN I/vLSIL (vagina low-grade squamous intraepithelial lesion), and VaIN II/III/vHSIL are 65.5%, 92.6%, and 98.5%, respectively. HPV-16 is the most common type in vaginal cancers (55.4%) and VaIN II/III/vHSIL (vaginal high-grade squamous intraepithelial lesion) (65.8%) [18]. These data are similar to the prevalence of HPV in high-grade squamous intraepithelial lesions (HSILs) of the cervix and cervical cancer [14]. The gold standard for the diagnosis of VaIN is colposcopy of the vagina [4]. The findings are classified in the same way as for the cervix, in accordance with the 2011 colposcopic terminology of the International Federation for Cervical Pathology and Colposcopy (IFCPC) [19].

The mean age of patients with VaIN II/III/vHSIL is between 54 and 60 years, and for vaginal cancer it is 63 years [20,21,22]. Almost half of VaIN III/vHSIL cases are associated with past cervical dypslasia. In the past decade, the diagnosis of VaIN has increased steadily [23]. The natural history of VaIN is thought to be similar to that of CIN [24]. Nevertheless, in some studies the time of progression from VaIN II/II/vHSIL I to SCC of the vagina has raised some concerns that there may be a different progression pathway in comparison with the progression of cervical HSIL to cervical cancer. VaIN III/vHSIL is considered to be a true premalignant disease and treatment is mandatory [24].

This means that rapid diagnosis of vHSIL is important in order to avoid progression to invasive cancer [4]. Cytology is a valid tool for the diagnosis of intraepithelial neoplasia and carcinoma of the cervix and vagina [25]. Only very limited data are available on the correct diagnosis of VaIN and vaginal cancer using cytology. There have only been a few studies correlating cytology findings with histological findings of VaIN. This study retrospectively correlated the results of cytology and HPV tests with the histological findings for biopsies obtained from the vagina.

The aim of this retrospective study was to compare the results of hrHPV and cytology co-testing with the histological findings in each case seen in the certified Dysplasia Unit. Additional findings were the indication for hysterectomy and a patient history of dysplasia in the lower genital tract. We are aiming to provide additional information about the natural course of VaIN and its risk factors in order to better diagnose it in future clinical routine.

## 2. Materials and Methods

Between January 2015 and January 2023, a total of 22,932 colposcopies were performed in the certified Dysplasia Unit at Erlangen University Hospital. A total of 18.397 women were excluded because no colposcopy of the vagina was performed. This is only performed in certain cases, e.g., history of VaIN or suspicious cytology and unsuspicious colposcopy of the cervix. Colposcopy of the vagina was performed in 4535 cases, and biopsies were taken in 384 cases; 105 cases of concomitant CIN were excluded. This left a total of 279 colposcopies in 209 women (Figure 1). Dysplasia units have been established nationally in Germany in accordance with the certification system of the German Cancer Society (Deutsche Krebsgesellschaft, DKG), the German Society for Gynecology and Obstetrics (Deutsche Gesellschaft für Gynäkologie und Geburtshilfe, DGGG), the Working Group on Gynecological Oncology (Arbeitsgemeinschaft Gynäkologische Onkologie, AGO), and the Working Group on Cervical Pathology and Colposcopy (Arbeitsgemeinschaft für Zervixpathologie und Kolposkopie, AGCPC). Dysplasia units cooperate with gynecological cancer centers in order to integrate in-patient healthcare facilities [26,27,28].

Abnormal cervical cytology findings were the most common reason for women being referred to the Dysplasia Unit. This study also included women who were referred for other reasons, such as lichen sclerosus or dysplasia of the vagina und vulva, in order to generate a large group of patients and obtain significant results [27]. Only women with a history of hysterectomy or women with normal cervical findings were included in the study. Women with concomitant CIN (LSIL and HSIL) or cervical cancer were excluded (Figure 1).

A conventional cytology of the cervix [29], an hrHPV test, and the application of 5% acetic acid to the cervix represent the standard of care in the unit and are carried out, in that order, on all women referred to the unit. In case of history of VaIN or abnormal cytology but normal colposcopy, Lugol’s iodine is applied to the vagina. Iodine-negative areas represent immature metaplasia, CIN, or low estrogen states. Complete iodine negativity—yellow staining in an area that has appeared strongly acetowhite—is highly suspicious for high-grade intraepithelial neoplasia in the vagina or cervix [4]. In the majority of cases, iodine-negative lesions represent the only hint for VaIN [30]. Only the cytology, hrHPV tests, and histology obtained at the same visit as the colposcopy were included in this study. Between 2015 and October 2018, the Hybrid Capture^®^ 2 test (HC2) was used to detect hrHPV (*n* = 2459). Since November 2018, hrHPV testing has been performed using the Abbott RealTime high-risk HPV assay on an Abbott m2000sp instrument (*n* = 1129).

The colposcopies were performed in standardized conditions using a Zeiss KSK 150 FC colposcope. General assessment was carried out in accordance with the 2011 IFCPC terminology [19]. The results for the cervical cytology were classified in accordance with the Bethesda nomenclature [31].

If there is a major finding or a lesion that is suspicious for invasion, a colposcopy-directed biopsy has to be taken from the most suspicious part of the lesion, using biopsy forceps (Seidl Biopsy Forceps ER076R; Aesculap AG, Tuttlingen, Germany) [32]. During the period of this retrospective analysis, the team in the Dysplasia Unit consisted of six colposcopists with various degrees of clinical experience and training. All data—such as colposcopic findings, cytology and hrHPV test results, histological outcomes, number of biopsies, type of transformation zone, and epidemiological outcomes—were recorded in a database for further research [27].

### 2.1. Statistical Analysis

In our analysis, we modeled histological findings by Pap findings and HPV status and evaluated the prediction accuracy by receiver operating characteristic (ROC) analysis.

For this, cytology grades were categorized into four different groups: benign (negative for intraepithelial lesion or malignancy (NILM), atypical squamous cells of undetermined significance (ASC-US)), low-grade squamous intraepithelial lesion (LSIL), HSIL+ (high-grade squamous intraepithelial lesion (HSIL), HSIL with features suspicious for invasion, squamous cell carcinoma), and unspecific (atypical glandular cells (AGC), endocervical favoring neoplasia, atypical squamous cells, cannot exclude HSIL (ASC-H)). Histological findings were dichotomized into low (benign, VaIN I/vHSIL) or high (VaIN II/vHSIL, VaIN III/vHSIL, vaginal cancer).

Using logistic regression models, dependencies were modeled between histology as a dependent variable and cytology grades as independent variables. To account for the fact that 279 observations were available from 208 patients, generalized estimating equations (GEEs) were used for estimation. The aim was to examine the predictive benefit of HPV as an additional independent variable in the model. Only those observations were therefore used for analysis in which HPV was available (271 observations from 201 patients). The data were then repeatedly (10,000 times) split into training data (size: four-fifths of all observations) and testing data (one-fifth of all observations) in order to estimate one model with cytology grading as the independent variable and then a second model with cytology grading and HPV as independent variables using the training data, and to predict the probability of belonging to the histological class “high” using the testing data. At each iteration, (ROC) analysis was performed for the relationship between the predictions of the models and the actual histological classes in the testing data. Finally, all ROC curves were averaged for both models using vertical averaging, and the *p* value was calculated for the difference between the areas under the curve (AUCs) in the two models, using the 95% confidence interval for all 10,000 AUC differences. Additionally, estimates for the odds ratios and their 95% confidence intervals and *p* values of the model were provided using cytology grading, and using HPV in addition to cytology grading, as independent variables obtained with the whole dataset used for training. The significance level was set to 0.05. All analyses were performed using the R statistics software program, version 4.2.2 statistics program (R Core Team (2022). R: A language and environment for statistical computing. R Foundation for Statistical Computing, Vienna, Austria) [33].

### 2.2. Ethical Approval

Approval for the study was obtained from the ethics committee of the Faculty of Medicine at Friedrich Alexander University of Erlangen–Nuremberg (reference number: 245_19 Bc). All procedures performed in this study involving human participants were in accordance with the ethical standards of the institutional and/or national research committee and with the 1964 Helsinki Declaration and its later amendments, or comparable ethical standards. For this retrospective study, no written informed consent was necessary.

## 3. Results

The median age of the women included was 49.8 years. Table 1 shows the data for cytology versus histology for the whole set of women (Table 1).

Of the total of 279 colposcopies, 153 had a history including hysterectomy and 126 had no hysterectomies. In the set of women with a history of hysterectomy, 11 of 13 (84.6%) with HSIL findings had VaIN III, and in the set of women without hysterectomy, three women in the group with HSIL findings (50%) had VaIN III. For VaIN II in the set of women with hysterectomy, half of the women were correctly diagnosed with HSIL, and in the set of women without a history of hysterectomy, the figure was 38.9% (Table 2 and Table 3); for results according to Munich III, see the Appendix A (see Appendix A).

The proportion of positive hrHPV tests increased with increasing severity of vaginal lesion (Table 4). The highest rate of positive hrHPV testing was in women with VaIN III (81.3%) (Table 4). A positive HPV test increased the risk for VaIN II/vHSIL+ by a factor of three (Table 5, Figure 2). A Pap finding of HSIL + significantly increased the risk for VaIN II/vHSIL, VaIN III/vHSIL, or carcinoma by a factor of 33.8, whereas LSIL or unspecific cytology findings had risk factors of 3.02 and 3.28 for VaIN II/vHSIL, VaIN III/vHSIL, or carcinoma (Table 5).

Previous hysterectomies for HPV-related CIN or carcinoma were noted for benign lesions, VaIN I/vLSIL, VaIN II/vHSIL, or VaIN III/vHSIL in 42.2%, 51.6%, 68.2%, and 63.6% of the patients, respectively (Table 6).

A history of preinvasive lesions in the lower genital tract was present in many patients with VaIN (199 of 279 cases, 71.3%) (Table 7).

## 4. Discussion

Vaginal intraepithelial neoplasia is a relatively rare preinvasive condition in gynecologic oncology [4]. It is diagnosed on the basis of abnormal cytology findings and/or a positive HPV test followed by a colposcopy-guided biopsy [1]. The prevalence of VaIN remains unclear, but in recent decades its incidence has risen due to the routine use of cytology tests in cervical cancer screening [34].

Unlike for colposcopy, there exist no strict rules for the colposcopy of the vagina and data are rare. The majority of VaINs are found in the upper third of the vagina [24]. A thickened epithelium seems to better predict a severe vaginal lesion, whereas a thin white epithelium better suggests a mild vaginal lesion [35]. Punctation is significantly associated with VaIN II/III/vHSIL, whereas mosaic and vascular patterns appear to be associated with VaIN III/vHSIL [24,30]. A micropapillary pattern was more commonly observed in women with VaIN I/vLSIL.

There have been numerous reports of a correlation between cytological findings and CIN/cervical cancer, but data on the correlation of VaIN and abnormal cytology are scarce [25]. This study presents data from the certified Dysplasia Unit at Erlangen University Hospital. The concordance rate was 60.6% (HSIL) for VaIN III/vHSIL and 53.7% for VAIN II/vHSIL. This is comparable to the retrospective study by Sopracordevole et al., including 87 patients with VAIN II/III/vHSIL [1]. In another retrospective study including more than 3000 women with VaIN of all grades, the sensitivity of cytology for detecting VaIN III/vHSIL was reported to be 75.6%. In that study, the cut-off for cytology in the sensitivity analysis was atypical squamous cells of undetermined significance (ASC-US) [23]. Co-testing with cytology and HPV tests increased the sensitivity to almost 99% [23]. In contrast, the sensitivity was only 42.1% for VaIN II/III/vHSIL in the retrospective study by Cong et al. [3].

An abnormal cytological result requires a subsequent colposcopy with intensive examination of the entire lower genital tract, including the vaginal walls and vault, as well as a biopsy of all suspicious lesions [1]. In addition to the known risk factors for HPV-induced neoplasia of the lower genital tract (e.g., smoking, immunosuppression, and multiple sexual partners), there are several risk factors that increase the likelihood specifically for VaIN: concomitant CIN, a history of CIN or VaIN, or a previous hysterectomy for HPV-related cervical invasive or preinvasive lesions [1,9,23,36].

Many studies reporting cytology results include cases of VaIN with concomitant CIN [3]. Cytological sampling of VaIN with concomitant CIN includes abnormal vaginal cells as well as abnormal cervical cells, which make up the largest proportion, as the cytology is normally taken from the cervix [3]. The sensitivity of a cytology is therefore higher for concomitant VaIN than for VaIN alone [3]. For this reason, all women with concomitant CIN were excluded from the present study. Only women with a normal colposcopy or benign histology for the cervix were included. Generally, the cervix is regarded as the most susceptible and most severe location for preinvasive neoplasia of the lower genital tract [37,38,39]. This means that examination of the vagina and vulva is potentially neglected during colposcopy [3]. In women with a previous hysterectomy, the cells in a cytology come from the vaginal wall, so that the sensitivity is higher for VaIN with a previous hysterectomy in comparison with women without a history of hysterectomy (VaIN II/vHSIL: 50% vs. 38.9%; VaIN III: 84.6% vs. 50% in the present study). An increased rate of sensitivity has also been reported in other studies, with a sensitivity of 69.5–82.9% for women with a previous hysterectomy in comparison with 59.5–77.0% for women without a previous hysterectomy [3,23].

Persistent infection has been found to be the major risk factor in the development of VaIN [4]. Human papillomavirus DNA is detected in 84–96% of cases of VaIN II/III/vHSIL. The most common type of HPV is type 16 [4,21,40,41]. Most VaIN III/vHSIL lesions are infected with HPV-16, but multiple infections are lowest in VaIN III/vHSIL [23]. The viral load correlates with the severity of the VaIN [42]. In the present study, the HPV infection rates increased with the severity of VaIN. More than 80% of patients with VaIN III/vHSIL were positive for HPV, while only 54.1% of those with VaIN I/vLSIL were positive. The rate of HPV was lower for vaginal cancer than for VaIN III/vHSIL. Most likely the low number of vaginal cancers (*n* = 3) is responsible for these misleading data. The number of vaginal cancers is too low to draw a conclusion from these data.

Another risk factor for VaIN is a history of hysterectomy for HPV-related CIN or cervical cancer. VaIN is reported to occur seven times more often in women with a history of HPV-related CIN or cervical cancer [43]. In a study by Coughlan et al., 123 women who had undergone primary hysterectomy for cervical cancer were evaluated for the effectiveness of vaginal cytology following hysterectomy. Twelve women were found to have developed cytological abnormalities at the routine follow-up examinations—seven with LSIL and five with HSIL. In six women, the examiners found positive colposcopic findings, with subsequent VaIN II/III/vHSIL in four cases and vaginal cancer in two cases [44]. In the present study, 68.2% of patients with VaIN II and 63.6% of those with VaIN III/vHSIL had a history of hysterectomy for HPV-related CIN or cervical cancer. There are no specific directive or screening programs for VaIN or vaginal cancer. Clinicians therefore appear to adopt the recommendations that have been published for abnormal cervical cytology [43]. Accurate data on the percentage of VaIN cases after hysterectomy are limited, as many women stop attending cytological and hrHPV testing after hysterectomy [3]. When there are abnormal cytological findings after hysterectomy, a careful examination of the whole vagina is necessary in order not to miss VaIN, especially in women with a history of hysterectomy for HPV-related CIN or cervical cancer.

Previous neoplasia in the lower genital tract is another important risk factor for VaIN. In the retrospective study by Ao et al., previous neoplasia in the lower genital tract was noted in 278 women (8.6%). The vast majority of the diagnoses were cervical lesions. There was a significant difference in the grades of previous cervical lesions between VaIN I, VaIN II/vHSIL, and VaIN III/vHSIL. The higher the grade of the previous CIN, the higher the grade of VaIN that was diagnosed [23]. In the present study, a history of (pre-)malignant disease in the lower genital tract was observed in 199 cases (71.3%). This comparatively large number of patients might be explained by the fact that many women who have undergone therapy for (pre-)malignant disease have follow-up examinations in the certified Dysplasia Unit or are referred back to it by their own gynecologists in case of an abnormal cytology. A history of CIN III was present in 29.6% of women with VaIN III/vHSIL, but only one woman with VaIN III/vHSIL had a history of CIN II. A history of CIN was much more frequent than a history of VaIN or vulvar intraepithelial neoplasia (VIN; CIN III, *n* = 56; VaIN III/vHSIL, *n* = 23; and VIN III, *n* = 12). The grade of previous CIN was also higher in women with/vHSIL, as reported in other studies.

The above data show that a history of HPV-related disease in the lower genital tract is an important risk factor for the development of VaIN II/III/vHSIL. In the study by Sopracordevole et al., 13 women were diagnosed with vHSIL after hysterectomy that was performed for non-HPV-related disease. However, 12 of the women had a history of HPV-related cervical lesions that had been treated conservatively before the hysterectomy [1]. These data show the importance of cytological follow-up in women with a history of hysterectomy for non-HPV-related disease when there are risk factors for VaIN [1]. The national S3 Guideline for prevention of cervical cancer recommends for HPV-positive women after hysterectomy an aftercare with cytology and an hrHPV test, as these women have an increased risk for VaIN [45].

### Strengths and Limitations

This study has certain limitations. Firstly, it is a retrospective study, and this may have limited the availability of some information, e.g., the reason for hysterectomy or a history of pre-malignancy. Secondly, interobserver variability cannot be ruled out. In addition, the cytological and histological findings were analyzed in the same department, in some cases by the same examiner. The cytologists were aware of the colposcopic appearance and therefore knew whether there was a suspicious lesion. This may have influenced the results. Hence, colposcopy of the vagina is not a standard procedure according to the German Cancer Early Detection Directive of cervical cancer (KFE-RL); it is not routinely performed in our dysplasia unit and therefore the majority if women were ruled out [46]. Therefore, we potentially have missed a number of VaIN cases. Thirdly, there were no control samples. The Dysplasia Unit in Erlangen is specialized for neoplasia and carcinoma in the lower genital tract. The probability of women having neoplasia or carcinoma is therefore higher than in the general population. In some cases, the medical records were incomplete. Due to the rarity of VaIN, women with an intact uterus were also included. In these cases, there was no choice but to collect the swabs from the cervix. To reduce potential bias, only women with normal colposcopic findings of the cervix and/or benign histology of the cervix were included, otherwise we would not have been able to differentiate between suspicious cytology of the cervix or vagina. Women with concomitant CIN or cervical cancer were excluded. Due to the challenging nature of taking a biopsy from the vagina, it is possible for VaIN II/III/vHSIL to be missed, with a biopsy being taken from the surrounding VaIN I/vLSIL instead.

## 5. Conclusions

VaIN is a rare premalignant disease of the lower genital tract, for which the main risk factor is HPV infection. The HPV detection rate increases with the severity of VaIN. An HPV co-test should be part of routine testing in women with a high risk of VaIN, such as women with a history of VaIN or hysterectomy for HPV-related disease. The detection rate for VaIN III/vHSIL was 50% after hysterectomy and 36.4% without hysterectomy. The detection rate is higher in women with a previous hysterectomy in comparison with women without a history of hysterectomy. In women with previous hysterectomies carried out due to HPV-related CIN or cervical cancer, the rates of VaIN II/III/vHSIL were higher in comparison with women with hysterectomies that were not related to HPV disease. Women who have risk factors for VaIN should undergo careful examination of the entire vaginal walls in order to rule out VaIN or vaginal cancer.

## Figures and Tables

**Figure 1 cancers-15-04633-f001:**
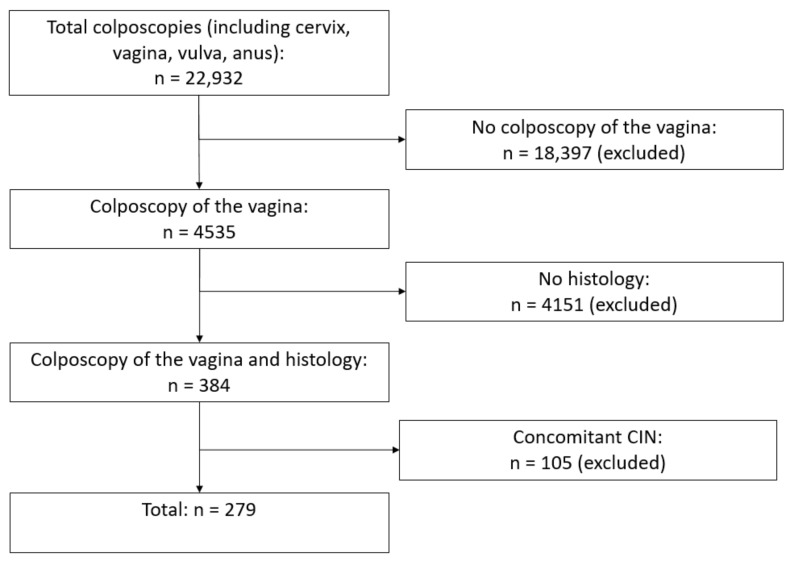
Flowchart.

**Figure 2 cancers-15-04633-f002:**
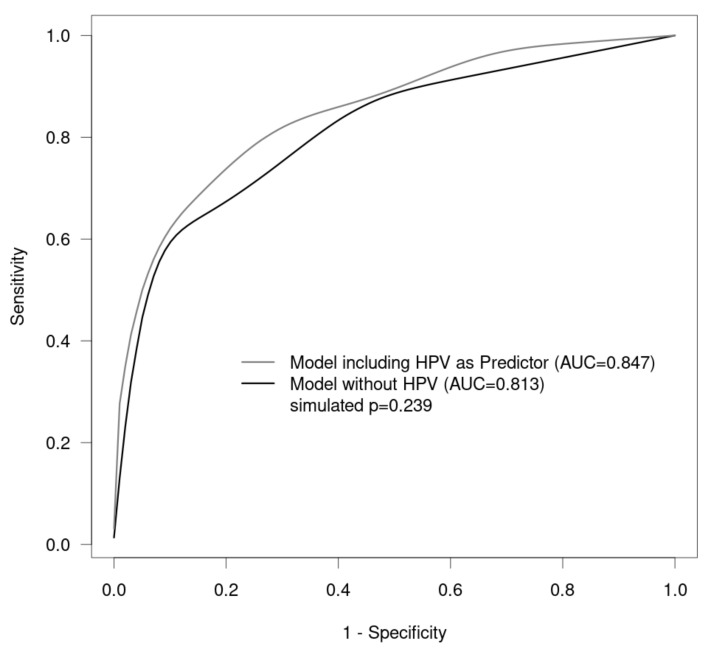
Receiver operating characteristic (ROC) curves for the models with and without HPV (risk for histology “high”).

**Table 1 cancers-15-04633-t001:** Cytology versus histology (*n* = 279).

Bethesda	Benign (*n* = 86)	VaIN I/vLSIL (*n* = 116)	VaIN II/vHSIL (*n* = 41)	VaIN III/vHSIL (*n* = 33)	Carcinoma (*n* = 3)
NILM (*n* = 107)	50 (46.7%)	51 (47.7%)	4 (3.7%)	2 (1.9%)	0
ASC-US (*n* = 17)	6 (35.3%)	8 (47.1%)	2 (11.8%)	1 (5.9%)	0
LSIL (*n* = 70)	17 (24.3%)	36 (51.4%)	11 (15.7%)	6 (8.6%)	0
HSIL (*n* = 57)	4 (7.0%)	11 (19.3%)	22 (38.6%)	20 (35.1%)	0
AGC, endocervical favoring neoplasia (*n* = 1)	1 (100%)	0	0	0	0
ASC-H (*n* = 24)	8 (33.3%)	10 (41.6%)	2 (8.3%)	3 (12.5%)	1 (4.2%)
HSIL with features suspicious for invasion (*n* = 1)	0	0	0	1 (100%)	0
Squamous cell carcinoma (*n* = 2)	0	0	0	0	2 (100%)

AGC, atypical glandular cells; ASC-H, atypical squamous cells, cannot exclude HSIL; ASC-US, atypical squamous cells of undetermined significance; HSIL, high-grade squamous intraepithelial lesion; LSIL, low-grade squamous intraepithelial lesion; NILM, negative for intraepithelial lesion or malignancy; VaIN, vaginal intraepithelial neoplasia.

**Table 2 cancers-15-04633-t002:** Cytology versus histology (after hysterectomy; *n* = 153).

Bethesda	Benign (*n* = 45)	VaIN I/vLSIL (*n* = 62)	VaIN II/vHSIL (*n* = 22)	VaIN III/vHSIL (*n* = 22)	Carcinoma (*n* = 2)
NILM (*n* = 62)	28 (45.2%)	29 (46.8%)	3 (4.8%)	2 (3.2%)1	0
ASC-US (*n* = 8)	3 (37.5%)	4 (50.0%)	0 (0%)	1 (12.5%)	0
LSIL (*n* = 33)	7 (21.2%)	19 (57.6%)	5 (15.2%)	2 (6.1%)	0
HSIL (*n* = 37)	1 (2.7%)	7 (18.9%)	13 (35.1%)	16 (43.2%)	0
AGC, endocervical favoring neoplasia (*n* = 0)	0	0	0	0	0
ASC-H (*n* = 12)	6 (50.0%)	3 (25%)	1 (8.3%)	1 (8.3%)	1 (8.3%)
HSIL with features suspicious for invasion (*n* = 0)	0	0	0	0	0
Squamous cell carcinoma (*n* = 1)	0	0	0	0	1 (100%)

AGC, atypical glandular cells; ASC-H, atypical squamous cells, cannot exclude HSIL; ASC-US, atypical squamous cells of undetermined significance; HSIL, high-grade squamous intraepithelial lesion; LSIL, low-grade squamous intraepithelial lesion; NILM, negative for intraepithelial lesion or malignancy; VaIN, vaginal intraepithelial neoplasia.

**Table 3 cancers-15-04633-t003:** Cytology versus histology (without hysterectomy; *n* = 126).

Bethesda	Benign (*n* = 41)	VaIN I/vLSIL (*n* = 54)	VaIN II/vHSIL (*n* = 19)	VaIN III/vHSIL (*n* = 11)	Carcinoma (*n* = 1)
NILM (*n* = 45)	22 (48.9%)	22 (48.9%)	1 (2,2%)	0	0
ASC-US (*n* = 9)	3 (33.3%)	4 (44.4%)	2 (22.2%)	0	0
LSIL (*n* = 37)	10 (27.0%)	17 (45.9%)	6 (16.2%)	4 (10.8%)	0
HSIL (*n* = 20)	3 (15.0%)	4 (20.0%)	9 (45.0%)	4 (20.0%)	0
AGC, endocervical favoring neoplasia (*n* = 1)	1 (100%)	0	0	0	0
ASC-H (*n* = 12)	2 (16.7%)	7 (58.3%)	1 (8.3%)	2 (16.7%)	0
HSIL with features suspicious for invasion (*n* = 1)	0	0 (0%)	0 (0%)	1 (100%)	0
Squamous cell carcinoma (*n* = 1)	0	0	0	0	1 (100%)

AGC, atypical glandular cells; ASC-H, atypical squamous cells, cannot exclude HSIL; ASC-US, atypical squamous cells of undetermined significance; HSIL, high-grade squamous intraepithelial lesion; LSIL, low-grade squamous intraepithelial lesion; NILM, negative for intraepithelial lesion or malignancy; VaIN, vaginal intraepithelial neoplasia.

**Table 4 cancers-15-04633-t004:** High-risk HPV status vs. vaginal histology (*n* = 270).

Histology	hrHPV-Positive	hrHPV-Negative
Benign (*n* = 84)	35 (41.7%)	49 (58.3%)
VaIN I/vLSIL (*n* = 111)	60 (54.1%)	51 (45.9%)
VaIN II/vHSIL (*n* = 40)	31 (77.5%)	9 (22.5%)
VaIN III/vHSIL (*n* = 32)	26 (81.3%)	6 (18.7%)
Carcinoma (*n* = 3)	2 (66.7%)	1 (33.3%)

hrHPV, high-risk human papillomavirus; VaIN, vaginal intraepithelial neoplasia.

**Table 5 cancers-15-04633-t005:** Results of a logistic regression model (GEE) with histological findings (low: benign, VaIN I vHSIL vs. high: VaIN II vHSIL, VaIN III vHSIL, vaginal cancer) modeled by Pap and HPV.

	Odds Ratio	95% Confidence Intervals	*p* Value
Cytology LSIL	3.02	1.18 to 7.7	0.021
Cytology HSIL+	33.80	12.74 to 89.68	<0.001
Cytology unspecific	3.28	1.14 to 9.42	0.028
HPV positive	2.99	1.51 to 5.93	0.002

HPV, human papillomavirus; HSIL, high-grade squamous intraepithelial lesion; LSIL, low-grade squamous intraepithelial lesion.

**Table 6 cancers-15-04633-t006:** Reason for previous hysterectomy.

Histology of Vagina (*n* = 153)	Hysterectomy for HPV-Related CIN or Cervical Cancer (*n* = 80)	Hysterectomy for Non-HPV-Related CIN or Cervical Cancer (*n* = 32)	Reason for Hysterectomy Unknown (*n* = 41)
Benign (*n* = 45)	19 (42.2%)	12 (26.7%)	14 (31.1%)
VaIN I/vLSIL (*n* = 62)	32 (51.6%)	14 (22.6%)	16 (25.8%)
VaIN II/vHSIL (*n* = 22)	15 (68.2%)	4 (18.2%)	3 (13.6%)
VaIN III/vHSIL (*n* = 22)	14 (63.6%)	1 (4.5%)	7 (31.8%)
Carcinoma (*n* = 2)	0	1 (50%)	1 (50%)

**Table 7 cancers-15-04633-t007:** History of (pre-)malignancy of the cervix, the vagina, and the vulva (*n* = 199).

	Benign (*n* = 50)	VaIN I/vLSIL (*n* = 90)	VaIN II/vHSIL (*n* = 31)	VaIN III/vHSIL (*n* = 27)	Carcinoma (*n* = 1)
CIN I/cLSIL (*n* = 10)	6 (12%)	2 (2.2%)	1 (3.2%)	1 (3.7%)	0
CIN II/cHSIL (*n* = 16)	2 (4%)	9 (10%)	4 (12.9%)	1 (3.7%)	0
CIN III/cHSIL (*n* = 56)	11 (22%)	26 (28.9%)	11 (35.5%)	8 (29.6%)	0
Cx-Ca. (*n* = 49)	17 (34%)	15 (16.7%)	7 (22.6%)	9 (33.3%)	1 (100%)
VaIN I/vLSIL (*n* = 5)	2 (4%)	2 (2.2%)	1 (3.2%)	0	0
VaIN II/vHSIL (*n* = 21)	3 (6%)	14 (15.6%)	3 (9.7%)	1 (3.7%)	0
VaIN III/vHSIL (*n* = 23)	5 (10%)	12 (13.3%)	1 (3.2%)	5 (18.5%)	0
Vaginal-Ca. (*n* = 2)	1 (2%)	0	1 (3.2%)	0	0
VIN I/vuLSIL (*n* = 0)	0	0	0	0	0
VIN II/vuHSIL (*n* = 1)	1 (2%)	0	0	0	0
VIN III/vuHSIL (*n* = 12)	1 (2%)	8 (8.9%)	2 (6.5%)	1 (3.7%)	0
Vulva-Ca. (*n* = 4)	1 (2%)	2 (2.2%)	0 (0%)	1 (3.7%)	0

VuHSIL = vulvar HSIL.

## Data Availability

The data supporting the findings of this study are available from the corresponding author upon reasonable request.

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
