# Peer review of "Cytology and HPV Co-Testing for Detection of Vaginal Intraepithelial Neoplasia: A Retrospective Study"

_cancers, 2023, doi:10.3390/cancers15184633_

Round 1

Reviewer 1 Report

General Overview:

The manuscript discusses important aspects of vaginal intraepithelial neoplasia (VaIN), its risk factors, and the correlation between abnormal cervical cytology and positive HPV test and histological findings of VaIN.

The manuscript provides valuable insights into the diagnosis of VaIN, which can be helpful for clinicians and researchers in the field.

Some suggestions for improvement include:

- In the abstract, please better define the purpose of the study.

- The introduction sets the context well and presents relevant information about the incidence, risk factors, and diagnostic methods for VaIN. In addition, please better define the objective of the study and the reasons for which it is carried out. Please do not repeat this at the end of the statistical analysis section.

- Why do the authors differentiate colposcopy of the vagina from colposcopy of the cervix? in all examinations carried out for altered screening tests (pap test or HPV test) both the cervix and the vaginal walls must be well observed, as reported in international recommendations.

- In the introduction section (56): a large retrospective case series reported a higher risk of progression for women with VaIN3 and for women with previous hysterectomy for cervical HPV-related disease. Please include this study (PMID: 27010135.)

- In the methods section (115-117): A conventional Pap smear of the cervix (using the Munich III nomenclature (standard in Germany) [26]), an hrHPV test, and application of 5% acetic acid to the cervix represent the standard of care in the unit and are carried out in that order on all women referred to the unit. Are pap smear or co-testing done (or repeated) in all patients? For what reason? Colposcopy is done in most cases after alteration of these tests. The patient should be monitored for this very reason. Please clarify. - In the methods section (123-124) Only Pap smears, hrHPV tests, and histology obtained  at the same visit as the colposcopy were included in this study. Therefore, it is not clear why the included patients undergo colposcopy. Haven't they already done these tests? - In the methods section (131): Bethesda nomenclature. Please correct. - In the results section, I recommend not duplicating data. Report in the results section which type of data is shown in the tables. Reporting too many results makes the text difficult to read. - Table 5. please define what the odds ratio is calculated for. - I can't find table 6 and 7 mentioned in the results. - In the discussion section (266): Only women with a normal colposcopy or benign histology were included. Cervical colposcopy? Benign cervical histology? there is confusion in the text for this.

- VaIN1 (LG-VaIN) should be regarded as the sole expression of the viral infection and not as a pathology. For this reason, VaIN1 is subjected to treatment very rarely and only in certain cases. The most interesting risk of progression is for VaIN2+. Please discuss this concept

Additional information:

-        It would be beneficial to provide more context on the significance of VaIN in the context of gynecologic oncology and its implications for patient management and care.

- The manuscript would benefit from more clarity and organization. Some sections seem repetitive (for example, tables and results, methods and strength or limitations, etc. ), and the flow could be improved to make the content more coherent.

1.     Reorganize the content to improve flow and coherence.

2.     Ensure that each section presents unique and relevant information, avoiding repetition.

3.     Break long sentences into shorter, more concise ones to enhance readability and clarity.

.

Author Response

Reviewer 1:

The manuscript discusses important aspects of vaginal intraepithelial neoplasia (VaIN), its risk factors, and the correlation between abnormal cervical cytology and positive HPV test and histological findings of VaIN.

The manuscript provides valuable insights into the diagnosis of VaIN, which can be helpful for clinicians and researchers in the field.

Dear Reviewer,

Thank you very much for your valid review. We have replied to it point by point below. We hope that you agree with the changes we made.

In the abstract, please better define the purpose of the study.

Important comment. We have added a sentence defining the purpose of the study.

The introduction sets the context well and presents relevant information about the incidence, risk factors, and diagnostic methods for VaIN. In addition, please better define the objective of the study and the reasons for which it is carried out. Please do not repeat this at the end of the statistical analysis section.

Thank you very much for this valid comment.

We have moved it from the end of the statistical analysis section to the end of introduction. We are aiming to provide additional information about the natural course of VaIN and its risk factors in order to better diagnose it in future clinical routine (line 119-124).

Why do the authors differentiate colposcopy of the vagina from colposcopy of the cervix? in all examinations carried out for altered screening tests (pap test or HPV test) both the cervix and the vaginal walls must be well observed, as reported in international recommendations

A visual inspection of the vagina is performed to all women. In case of suspicious lesion or high risk for VaIN the vagina is stained with lugol’s iodine. In Germany there exist strict regulations how to perform a colposcopy according to the  German Cancer Early Detection Directive of cervical cancer [KFE-RL]. The visual inspection but not the routine examination lugols iodine and / or kolposcopy of the vagina is not mandatory in clinical routine,

In the introduction section (56): a large retrospective case series reported a higher risk of progression for women with VaIN3 and for women with previous hysterectomy for cervical HPV-related disease. Please include this study (PMID: 27010135.)

We have included this study as requested by you. It is literature number 10.

In the methods section (115-117): A conventional Pap smear of the cervix (using the Munich III nomenclature (standard in Germany) [26]), an hrHPV test, and application of 5% acetic acid to the cervix represent the standard of care in the unit and are carried out in that order on all women referred to the unit. Are pap smear or co-testing done (or repeated) in all patients? For what reason? Colposcopy is done in most cases after alteration of these tests. The patient should be monitored for this very reason. Please clarify. - In the methods section (123-124) Only Pap smears, hrHPV tests, and histology obtained  at the same visit as the colposcopy were included in this study. Therefore, it is not clear why the included patients undergo colposcopy. Haven't they already done these tests? - In the methods section (131): Bethesda nomenclature. Please correct. - In the results section, I recommend not duplicating data. Report in the results section which type of data is shown in the tables. Reporting too many results makes the text difficult to read.

Thank you for your important comments. There are strict recommendations on how to perform a colposcopy according to the German Cancer Early Detection Directive of cervical cancer [KFE-RL] as described above. In Germany, we are inforced by law to adhere to these recommendations.

The main task in our dyplasia unit is the cervical cancer screening assessment. Women with abnormal cytology findings during cervical cancer screening are sent to our unit. In order to reevaluate the abnormal external cytology an in-house cytology is done. This is done routinely to all women.

Pap smears, hrHPV tests, and histology are obtained  at the same visit as the colposcopy. This is done in order to assess the ex-house cytology and hrHPV-test. In cases of low-grade cytology there are upt ot three month between primary screening and colposcopy in our unit. Therefore cytology findings may have change in the meantime and we repeat the cytology in our unit.

We have deleted repeating data.

Table 5. please define what the odds ratio is calculated for. 

The legend of the table was specified in order to clarify the statistics of this manuscript.

I can't find table 6 and 7 mentioned in the results.

We have added these two tables.

 In the discussion section (266): Only women with a normal colposcopy or benign histology were included. Cervical colposcopy? Benign cervical histology? there is confusion in the text for this.

Yes, we mean colposcopy and histology of the cervix. Otherwise we would not be able to differentiate between suspicious cytology of cervix or vagina.

A VaIN1 (LG-VaIN) should be regarded as the sole expression of the viral infection and not as a pathology. For this reason, VaIN1 is subjected to treatment very rarely and only in certain cases. The most interesting risk of progression is for VaIN2+. Please discuss this concept

This is a very important comment. We have already adressed this in the text. We have specified this in the introduction section (line 68-70).

It would be beneficial to provide more context on the significance of VaIN in the context of gynecologic oncology and its implications for patient management and care.

We have added some data regarding the natural course of VaIN and its implications for treatment in the introduction section.

The manuscript would benefit from more clarity and organization. Some sections seem repetitive (for example, tables and results, methods and strength or limitations, etc. ), and the flow could be improved to make the content more coherent

Thank you for pointing out this valid information. We have shortened the manuscript, especially the results sections. We deleted lines 224-232 as it included repetitive data of table 1.  

Reorganize the content to improve flow and coherence.

We did this for introduction, results and strenght and limitations section. Also discussions was rephrased.

      Ensure that each section presents unique and relevant information, avoiding repetition.

We tried to reorganize each section trying to avoid repetition. We added new information as requested by the reviewers in the introduction (line 58-74) and deleted the lines 91-101. Material and methods was partly rephrased (lines 150-169). This was also done for results (line 224-232) and discussion (e.g. 295-301 or 340-349)

      Break long sentences into shorter, more concise ones to enhance readability and clarity.

The sentences were shortened were applicable.

Reviewer 2 Report

I have read with great interest this paper by Stuebs and colleagues.

As well highlighted by the Authors, VaIN is a rare entity. Its natural history, clinical characteristics, diagnosis and treatment strategies need to be studied. Therefore, papers exploring novel insights on this condition are welcome.

In this paper the Authors aim to correlate the results of pap smear and HPV testing with histological findings in women with VaIN.

The paper is interesting, intelligible and well written but some minor issues needs to be assessed:

-          Introduction (lines 70-74) the LAST terminology for VaIN classification should be mentioned since it is widely used.

-          Lines 84-85 “there are no screening programs for VaIN ad vaginal cancer”. This is true but many cases of VaIN are diagnosed inside the national cervical cancer screening programs (thanks to the use of HPV or pap smears). Thus, this sentence is misleading and should be removed.

-          The introduction section is too long and could be shortened (e.g. removing lines 75-80).

-          Materials and methods: among all the 22932 colposcopies performed in the study period, only 4535 cases received a colposcopic evaluation of the vagina.

Why in 18397 cases the colposcopy of the vagina was not performed?

In my opinion this is not a “methodological” bias, but it is a “conceptual” bias and represents the main limit of the study, thus should be widely discussed in the proper section (“strengths and limitations”) of the manuscript.
In our daily practice we perform a complete evaluation of cervix and vagina in every woman who refers to our lower genital tract disease department. It is widely demonstrated that a complete vaginal evaluation during the colposcopy is mandatory in every woman (cases of CIN and concomitant VaIN are commonly observed). Avoiding the colposcopy of the vagina could lead to misdiagnosis of a huge amount of VaIN. Furthermore, Lugol’s iodine (line 118) should be routinely used since in many cases VaIN can present only as iodine-negative lesions (please see and cite doi: 10.1097/CEJ.0000000000000287).

All these aspects should be properly discussed by the authors.

-          Line 133: “only major colposcopic findings or lesions suspicious for invasion underwent a biopsy”. Even in this case this approach is not properly correct. The correlation between colposcopic features and histology of vaginal biopsy is poorly described in the literature. However, the few studies available (e.g. doi: 10.1111/j.1447-0756.2009.01108.x ; doi: 10.1097/LGT.0b013e318237ec82 ; doi: 10.1097/CEJ.0000000000000287) highlights the poor correlation between colposcopy and vaginal biopsy concluding that the severity of VaIN cannot be predicted by colposcopy pattern.

I suggest to add a paragraph in the discussion section to properly asses these issues about vaginal colposcopy.

-          Results: lines 184-193 can be removed since the results are clearly showed in Table 1

-          Results Table 4: It is interesting to observe the decrease of HR-HPV positivity in vaginal cancers compared to HG-VaIN. An explanation of this findings should be provided by the authors in the discussion section.

-          Table 6 and Table 7 (cited in line 231 and 233 respectively) are not shown in the manuscript.

-          It should be very interesting to observe the rate of HPV positivity after treatment of VaIN. Do the authors have such data of follow up? If these data are available, I suggest adding a proper paragraph and table showing the results.

-          Conclusion: The Authors aim to correlate the results of pap smear and HPV testing with histological findings in women with VaIN. As shown, the HPV detection rate increase with the severity of VaIN. The Authors should argue on how this finding could improve the usual clinical practice (e.g in your opinion a cotest should be routinely proposed in high-risk set of women such as those who underwent hysterectomy for HPV-related disease?)

english quality is quite good

Author Response

Dear Reviewer,

Thank you very much for your valid review. We have replied to it point by point below. We hope that you agree with the changes we made.

Introduction (lines 70-74) the LAST terminology for VaIN classification should be mentioned since it is widely used.

Thank you for this important comment. We have added the LAST terminology (line 65-74).

Lines 84-85 “there are no screening programs for VaIN ad vaginal cancer”. This is true but many cases of VaIN are diagnosed inside the national cervical cancer screening programs (thanks to the use of HPV or pap smears). Thus, this sentence is misleading and should be removed.

Your comment is true we therefore removed the sentence.

The introduction section is too long and could be shortened (e.g. removing lines 75-80)

We agree with you and have removed the lines 75-80.

Materials and methods: among all the 22932 colposcopies performed in the study period, only 4535 cases received a colposcopic evaluation of the vagina.

Why in 18397 cases the colposcopy of the vagina was not performed?

In my opinion this is not a “methodological” bias, but it is a “conceptual” bias and represents the main limit of the study, thus should be widely discussed in the proper section (“strengths and limitations”) of the manuscript.
In our daily practice we perform a complete evaluation of cervix and vagina in every woman who refers to our lower genital tract disease department. It is widely demonstrated that a complete vaginal evaluation during the colposcopy is mandatory in every woman (cases of CIN and concomitant VaIN are commonly observed). Avoiding the colposcopy of the vagina could lead to misdiagnosis of a huge amount of VaIN. Furthermore, Lugol’s iodine (line 118) should be routinely used since in many cases VaIN can present only as iodine-negative lesions (please see and cite doi: 10.1097/CEJ.0000000000000287).

All these aspects should be properly discussed by the authors.

18.397 women were excluded because no colposcopy of the vagina was performed. This is only done in certain cases e.g. history of VaIN or suspicious cytology and unsuspicious colposcopy of the cervix.

 A visual inspection of the vagina is performed to all women. In case of suspicious lesion or high risk for VaIN the vagina is stained with lugol’s iodine. In Germany there exist strict regulations how to perform a colposcopy according to the German Cancer Early Detection Directive of cervical cancer [KFE-RL]. The visual inspection but not the routine examination lugols iodine of the vagina is not mandatory.

We have cited the publication of Sopracordevole et al.

 line 133: “only major colposcopic findings or lesions suspicious for invasion underwent a biopsy”. Even in this case this approach is not properly correct. The correlation between colposcopic features and histology of vaginal biopsy is poorly described in the literature. However, the few studies available (e.g. doi: 10.1111/j.1447-0756.2009.01108.x ; doi: 10.1097/LGT.0b013e318237ec82 ; doi: 10.1097/CEJ.0000000000000287) highlights the poor correlation between colposcopy and vaginal biopsy concluding that the severity of VaIN cannot be predicted by colposcopy pattern.

I suggest to add a paragraph in the discussion section to properly asses these issues about vaginal colposcopy.

We totally agree, that colposcopy of the vagina is very challenging. Literature is rare. Therefore we are very grateful for the literature provided by you and have added a paragraph in the discussion (line 296-302)

Results: lines 184-193 can be removed since the results are clearly showed in Table 1

We have deleted that part of the manuscript and added a sentences introducing table 1 (line 224-233).

Results Table 4: It is interesting to observe the decrease of HR-HPV positivity in vaginal cancers compared to HG-VaIN. An explanation of this findings should be provided by the authors in the discussion section.

In our opinion the number of vaginal cancer (n=3) is too low to draw a conclusion from this data, but for reason of completeness we wanted to show the data.

Table 6 and Table 7 (cited in line 231 and 233 respectively) are not shown in the manuscript.

This is a very important notice. We have added these two tables.

 It should be very interesting to observe the rate of HPV positivity after treatment of VaIN. Do the authors have such data of follow up? If these data are available, I suggest adding a proper paragraph and table showing the results

Unfortunately we do not have these data. We agree it would be very interesting to analyze and discuss these data.

Conclusion: The Authors aim to correlate the results of pap smear and HPV testing with histological findings in women with VaIN. As shown, the HPV detection rate increase with the severity of VaIN. The Authors should argue on how this finding could improve the usual clinical practice (e.g in your opinion a cotest should be routinely proposed in high-risk set of women such as those who underwent hysterectomy for HPV-related disease?)

Very important point. In our opinion, the most effective would be a HPV co-testing for women with a history of VaIN and after hysterectomy for HPV related disease.  The national S3-Guideline for prevention of cervical cancer recommends for HPV-positive women after hysterectomy an aftercare with an cytology and a hrHPV-Test as these women have an increased risk for VaIN (line 391-393)

Reviewer 3 Report

The presented manuscript is written by a group of German researchers and it describes the correlation between cytology (and to some extent, also hrHPV test) results and vaginal histological findings among colposcopy patients with no cervical abnormalities. Due to the rarity of vaginal LSIL/HSIL (VaIN1-3) lesions and thus, respective literature, this manuscript is of certain interest especially for colposcopists. However, there are some major (and minor) issues that should be reconsidered before publication.

1. First and foremost, cytological results are primarily reported (especially in the tables) using Munich III nomenclature. For most of the international readers this is too complicated and, thus, reporting of the cytology should be uniformly done following the more commonly used Bethesda System throughout the paper. Tables representing the results according to the Munich III nomenclature could be given as supplementary material, if considered necessary.

2. Authors start the Introduction (and the Discussion and the Conclusions) by defining VaIN as "a rare premalignant disease". This is probably bit old-fashioned, as VaIN1 is likely to be mostly infectious and benign in nature even though the natural course of VaIN as whole is poorly understood. A few sentences on this subject with adequate references would be required. Moreover, is there any possibility that the authors could consider using terms vaginal LSIL (or vLSIL) instead of VaIN1 and vaginal HSIL (or vHSIL) instead of VaIN2-3 at least in the text to simplify the text? If so, also the heading should be edited correspondingly.

3. Related to the previous comment, the authors state in page 2 line 86 that "rapid diagnosis of vaginal HSIL is important in order to avoid progression to invasive cancer" - could you please add here some data on the natural course of vaginal HSIL to the readers?

4. In the Materials and Methods the authors adequately describe the study data. However, it is not explained why the colposcopy of the vagina was performed only for about 20% of more than 22,000 women originally having a colposcopy. Further, the reason for vaginal biopsies is not explained. It definitely should be mentioned, as the interpretation of the results is somewhat different if all the vaginal biopsies were taken in case of major/minor lesions compared to biopsies taken for any reason including random biopsies.

5. The Statistical Analysis section mostly describes the ROC analysis. Please, clarify this in the beginning of the section. Further, the aim of the study should rather be presented in the end of the Introduction than in the end of Statistical Analysis section.

6. Related to the comment 1: the Results section is difficult to follow due to uncommon/outdated classifications used for cytology and histology. Further, please make sure to use precise terminology in any case. For example, the sentence starting on line 184 (page 5) could be changed into: "In 65 colposcopies on women with normal cytology (negative for intraepithelial lesion or malignancy, NILM) at colposcopy, vaginal histology was benign in 44.6% and showed low-grade squamous intraepithelial lesion (LSIL) in 50.8%." etc. Further, it would be more elegant not to repeat the same figures in tables and text, for which I suggest, that the more detailed figures are given in the tables and only the figures/proportions for histologically benign/LSIL vs. HSIL+ cutoffs (i.e. benign/VaIN1 vs. VaIN2+) reported within the text, as this is clinically the most relevant distribution.

7. The sentence on the line 199 (page 5) starts with "Among the 279 patients, 153 had a history including hysterectomy...". However, should it be instead something like "Of the total 279 colposcopies, 153 were performed for women with previous hysterectomy and 126 for women without hysterectomy." as the number of women was previously reported to be 209 (page 2, line 97)?

8. Again, to avoid repeating the same information in the text and tables, I suggest the sentence starting at line 216 (page 6) to be shortened into: "The proportion of positive hrHPV tests increased with increasing severity of vaginal lesion (Table 4)." Moreover, I would not refer to the Figure 2 with the sentence starting from line 219 (page 6) as they do not seem related to me.

9. Please, remove the mention on Table 7. from the line 233 (page 7) as there doesn't seem to be such Table presented.

10. The Discussion is in most highly adequate. Only the second paragraph on page 9 (lines 317-324) seems to be in wrong place and should be either moved to the introduction or removed. Further, there are two almost identical phrases (starting at lines 317 and 319) that should be condensed to avoid repetition.

11. Concerning the Strengths and limitations: please, specify which information was limited due to the retrospective study design - the reasons for biopsies, colposcopic diagnoses etc.? Could some of the text here to be moved into Materials and methods?

12. As already mentioned above, the Tables 1-3 would be more informatic to most of the readers if condensed in a way that figures and proportions would be given only for cytological categories NILM, ASC-US, LSIL, ASC-H, HSIL+ and AGC. Results by Munich III classification could be given for example in supplementary files, if necessary. Further, could a more accurate title for the Table 4. to be e.g. "High-risk HPV status vs. vaginal histology"? Legends for Table 5. and Figure 2. are too unspecified to understand without reading the text; please clarify the cytological/histological cutoffs/comparison groups etc. where necessary. I suppose the font in the "Table 5." (line 224 on page 6) is in cursive by mistake?

Author Response

Dear Reviewer,

Thank you very much for your valid review. We have replied to it point by point in the word document attatched. We hope that you agree with the changes we made.

Round 2

Reviewer 3 Report

Thank you for your work in revising the manuscript. It has improved significantly. I pointed out just a few typos you might want to correct prior to the manuscript is finalized for publication:

On Page 2, line 68 the it is written that: "LSIL are regarded as Proliferation of squamous..." - should it rather be "proliferation" in lower case and either "LSILs are" or "LSIL is"?

Further, on page 4, lines 165-171 you write "To do so, cytology grades were categorized into four different groups: benign (, negative for intraepithelial lesion or malignancy (NILM), , atypical squamous cells of undetermined significance (ASC-US)), low-grade squamous intraepithelial lesion(LSIL) , HSIL+ (high-grade squamous intraepithelial lesion HSIL, HSIL with features suspicious for invasion, Squamous cell carcinoma , and unspecific (atypical glandular cells (AGC), endocervical favoring neoplasia, atypical squamous cells, cannot exclude HSIL (A SC-H))." and I suppose there are some excess commas and spaces within this. Please, correct these.

Otherwise I don't have anything else to comment and I sincerely recommend the manuscript for publication.

Author Response

Dear Reviwer, 

thank you very much again. We have corrected the typos pointed out by you. 

Best Regards

F. Stübs